# Residues and Safety Assessment of Cyantraniliprole and Indoxacarb in Wild Garlic (*Allium vineale*)

**DOI:** 10.3390/toxics11030219

**Published:** 2023-02-25

**Authors:** Syed Wasim Sardar, Jeong Yoon Choi, Yeong Ju Jo, Abd Elaziz Sulieman Ahmed Ishag, Min-woo Kim, Hun Ju Ham

**Affiliations:** 1Department of Biological Environment, Kangwon National University, Chuncheon 24341, Republic of Korea; 2Department of Crop Protection, University of Khartoum, Khartoum North, Shambat 13314, Sudan; 3Department of Residues and Contamination, Ministry of Food and Drug Safety, Cheongju-si 28159, Republic of Korea; 4Environmentally Friendly Agricultural Products Safety Center, Kangwon National University, Chuncheon 24341, Republic of Korea

**Keywords:** UPLC-MS/MS, insecticides, MRLs, PHI, safety assessment

## Abstract

In this study, the residual behavior and safety of cyantraniliprole and indoxacarb applied to wild garlic (*Allium vineale*) were investigated. Samples were harvested after treatments of 0, 3, 7, and 14 days, then were prepared and extracted following the QuEChERS method and analyzed by UPLC-MS/MS. The linearity (*R*^2^ ≥ 0.99) of the calibration curves was excellent for both compounds. The average recoveries of cyantraniliprole and indoxacarb at two spiking concentrations (0.01 and 0.1 mg/kg) ranged from 94.2% to 111.4%. The relative standard deviation value was below 10%. The initial concentrations of cyantraniliprole and indoxacarb in wild garlic were degraded to 75% and 93% after seven days. The average half-lives were 1.83 and 1.14 days for cyantraniliprole and indoxacarb, respectively. The preharvest intervals (PHIs) for the two pesticides in wild garlic are recommended as two treatments seven days before harvest. The safety assessment data indicated that the percent acceptable daily intakes of cyantraniliprole and indoxacarb were 0.3 × 10^−4^% and 6.7 × 10^−2^%, respectively, in wild garlic. The theoretical maximum daily intake value of cyantraniliprole was 9.80%, and that of indoxacarb was 60.54%. Both compounds’ residues in wild garlic pose low health risks to consumers. The findings of the current investigation provide essential data for the safe use of cyantraniliprole and indoxacarb in wild garlic.

## 1. Introduction

Crops grown on limited land areas or providing relatively low gross revenues are categorized as minor crops [1]. If crops do not provide sufficient financial payback, pesticide companies are unwilling to undertake expensive research to register pesticides for them; there will therefore be fewer options for the protection of such crops against pests [2]. In such cases, growers who cultivate minor crops apply non-registered pesticides to control pests [3]. The presence of non-registered pesticide residues in crops is of significant concern to consumers due to their health risks [4]. In addition, leafy vegetables cannot be peeled or trimmed, and their surface areas are large. The risk of pesticide residues being retained on these vegetables is relatively higher than for other crops [5].

Wild garlic (*A. vineale*) belongs to the family Amaryllidaceae and is mainly found in the Northern Hemisphere, in Europe, China, and Korea [6]. In the Republic of Korea, wild garlic’s green leaves and bulbs are used as vegetables as well as for pickling, making kimchi, and in traditional medicine [7,8]. As the consumption of wild garlic has risen in the Republic of Korea, growers have begun growing the crop in greenhouses. Nevertheless, farmers have problems managing various pests (e.g., lepidopteran and sucking insects) due to the limited number of registered pesticides for wild garlic. In the current study, cyantraniliprole and indoxacarb were chosen as target insecticides for wild garlic. Cyantraniliprole (CAS RN, 736994-63-1) is a comparatively new systemic diamide insecticide that specifically controls lepidopteran pests and sap-sucking insects in agriculture [8,9]. It is a promising insecticide because of its distinctive mechanism of action that involves activating ryanodine receptors, which are important for muscle contraction. Activation of these receptors in insects causes the uncontrolled release of calcium from muscle cells, leading to contraction and paralysis and, ultimately, death [10,11]. In the Republic of Korea, cyantraniliprole has already been used for lepidopteran control on several vegetables and fruits [12]. Indoxacarb (CAS RN, 144171-61-9) is a broad-spectrum new oxadiazine-type insecticide registered against lepidopteran and sucking insects for various crops, fruits, and vegetables. It generally acts on the insect nervous system by blocking the movement of sodium ions in nerve cells, causing paralysis and, eventually, the death of pests [8,13]. Numerous studies have investigated the residue behavior and dissipation patterns of cyantraniliprole and indoxacarb in several crops [12,14,15,16].

However, no studies have been conducted on cyantraniliprole and indoxacarb dissipation and residual behavior in wild garlic. Furthermore, there is no maximum residue limit (MRL) for cyantraniliprole in wild garlic in the Republic of Korea. Therefore, the aim of this study was (1) to explore the dissipation and residue behavior of cyantraniliprole and indoxacarb in wild garlic under greenhouse conditions, (2) to obtain primary data on the safe use of cyantraniliprole and indoxacarb in wild garlic, (3) and to estimate the percent acceptable daily intakes (%ADIs) and theoretical maximum daily intakes (TMDIs) for Korean consumers using the obtained residual data, MRLs, and food factors and comparing them with the acceptable daily intakes (ADIs) to assess the human health risks.

## 2. Materials and Methods

### 2.1. Standards and Reagents

Cyantraniliprole and indoxacarb standards (>94.0% purity) were supplied by Toronto Research Chemicals and Aquastandard, respectively (Appendix A). HPLC-grade solvents, including water, acetonitrile, and methanol, were obtained from Thermo Fisher Scientific (Waltham, MA, USA). QuEChERS kits (EN 15662) were purchased from Phenomenex, CA, USA, and d-SPE PSA kits (150 mg magnesium sulfate and PSA 50 mg) were provided by Bekolut (Germany). Individual working solutions of both pesticides were made in acetonitrile.

### 2.2. Field Experiments

Field experiments for cyantraniliprole and indoxacarb in wild garlic were conducted in a greenhouse during the 2022 season in Goesan City, the Republic of Korea. The field area was divided as previously described by Sardar et al. [17]. The insecticides cyantraniliprole 5% EC and indoxacarb 10% WP were diluted 2000 times and applied using an automatic sprayer (Appendix A). The applications were made twice at 7-day intervals. After the second spray, wild garlic samples were harvested from each subplot following the Korean Rural Development Administration (RDA) method [18], and the samples were directly shipped to the laboratory for examination.

All plants used in the present study complied with the IUCN Policy Statement on Research Involving Species at Risk of Extinction and the Convention on the Trade in Endangered Species of Wild Fauna and Flora.

### 2.3. Sample Pretreatment and Analysis

The QuEChERS method was utilized to extract cyantraniliprole and indoxacarb residues from wild garlic. Briefly, triplicate samples (10 g each) were accurately weighed, placed in a 50 mL conical tube, and extracted by applying the procedure previously described by Sardar et al. [17]. The samples were vigorously shaken at 1500 rpm in an automatic shaker for 10 min. Subsequently, the QuEChERS kit was added, shaken again for 2 min, followed by centrifugation at 4000 rpm for 10 min to obtain the supernatant. The supernatant was collected and filtered (0.2 μm membrane) in a glass vial (1.5 mL) for analysis.

UPLC coupled with MS/MS (details in Appendix A) was used for cyantraniliprole and indoxacarb residue analysis in wild garlic. Mobile phases A (water) and B (methanol) acidified with 5 mM ammonium format and 0.1% formic acid were used. The injection volume was 1.0 µL. The target analytes were separated using a Unison UK-C18 column (Imtakt, Portland, OR, USA) with a gradient elution described and presented in the Appendix A). 

MS/MS analysis was performed with a TSQ Quantum Ultra (Thermo Science, Waltham, MA, USA) equipped with electrospray ionization (ESI). Both analytes were detected in positive ion mode. The optimized parameters and selected reaction monitoring (SRM) conditions are given in Appendix A. 

### 2.4. Method Validation

Different performance parameters were assessed to confirm the analytical method’s suitability. Six-point matrix-matched calibration curves were constructed for the quantification of the target analytes. The linearity was evaluated by regression coefficient (R^2^) values. The accuracy and precision of the method were determined by recovery (acceptable range: 70–120%) and reproducibility (n = 3) experiments. The recoveries were obtained by fortification of the target analytes at two concentrations (0.01 and 0.1 mg/L) in control samples (10 g). The repeatability was assessed by calculating the triplicate samples’ relative standard deviations (RSDs). The limit of detection (LOD) was calculated as the lowest concentration of target analytes generating a peak with a signal-to-noise ratio (S/N) of 3, while the limit of quantification (LOQ) was calculated following Sardar et al. [17], considering the sample amount, the volume of extraction solvent, the dilution factor, the injection volume, and the LOD.

### 2.5. Initial Residue Calculations

The initial residues of cyantraniliprole and indoxacarb deposition in wild garlic were calculated from day 0 [17,19].

### 2.6. Half-Life Calculations

The half-lives of cyantraniliprole and indoxacarb in wild garlic were calculated using the equation described by Sardar et al. [17], following the first-order kinetics model. 

### 2.7. Safety Assessments

The safety assessments in terms of percent acceptable daily intakes (%ADIs) of the target pesticides in wild garlic were calculated by comparing estimated daily intakes (EDIs) and acceptable daily intakes (ADIs). Averages of agricultural products consumed per person in one day and pesticide concentrations were used to estimate EDIs. Additionally, each pesticide’s theoretical maximum daily intakes (TMDIs) were estimated utilizing average body weight (60 kg) and the MRL values for the pesticides. The TMDIs were calculated according to Kim et al. [20].

## 3. Results and Discussion

### 3.1. Method Validation

The cyantraniliprole and indoxacarb analytical methods were evaluated using linearity, LOD, LOQ accuracy, and precision. The linearity of the six-point matrix matched the calibration curve for cyantraniliprole (0.005 to 0.1 mg/kg), and the results for indoxacarb (0.002 to 0.1 mg/kg) were good, with a correlation coefficient (R^2^) greater than 0.99 (Appendix A). The recovery values for cyantraniliprole and indoxacarb at 0.01 and 0.1 mg/kg spiking levels ranged from 91.2 to 105.7% and 103.6 to 116.1%, respectively, in wild garlic (Appendix A). The reproducibility of the analytical method (n  =  3) was less than 10% for both analytes. The LOD of cyantraniliprole was 0.005 mg/kg, and that of indoxacarb was 0.002 mg/kg. The LOQ for both analytes was estimated to be 0.01 mg/kg (Table 1). No interference peaks from co-eluting substances were observed near the peaks of the analytes in spiked and unspiked blank samples, which indicates the accuracy of the analytical method.

### 3.2. Initial Depositions and Dissipations of Cyantraniliprole and Indoxacarb 

The initial depositions of cyantraniliprole and indoxacarb are the estimated theoretical amounts of these insecticides that could settle on wild garlic. The initial depositions were calculated from 0-day residues (residues detected in samples collected 2 h after pesticide application), taking into account the AI (active ingredient) percentages in the formulations, the formulation types, and the dilution factors [17,19]. The initial residues (mg/kg) of cyantraniliprole and indoxacarb in wild garlic were 0.4 and 0.7 mg/kg, respectively, as shown in Table 2. Indoxacarb showed initial residues slightly higher than those of cyantraniliprole. The higher residues of indoxacarb could not be correlated with differences in plant growth, environmental conditions, or application rate, as these were similar for both pesticides throughout the experimental trials. The active ingredients used in this study were cyantraniliprole 5% EC and indoxacarb 10% WP. Indoxacarb’s higher initial residues were attributed to the high amount of active ingredient in the commercial formulation. These findings agree with the results of previous studies that reported that the initial residues of pesticides in crops depend on various factors (formulation, active ingredient, type of crop, dilution factor, and physicochemical properties of pesticides) [3,17,21]. In the case of indoxacarb, the log P value was higher than that of cyantraniliprole, which is considered to easily penetrate the crop, leading to higher residues [22].

Evaluations of pesticide dissipation and half-lives offer vital indexes to determine the behavior of pesticides in crops. The half-lives of cyantraniliprole and indoxacarb were obtained by first-order kinetics. The dissipations of cyantraniliprole and indoxacarb in wild garlic are shown in Figure 1. The residues of cyantraniliprole and indoxacarb gradually decreased over time, as the duration from the pesticides’ last spraying to the harvest date increased. The residues of cyantraniliprole and indoxacarb in wild garlic were dissipated by 75% and 93% after seven days, with respective half-lives (days) of 1.83 and 1.14. Cyantraniliprole and indoxacarb half-lives in wild garlic were less than a week. Malhat et al. [14] reported a similar result for cyantraniliprole half-life in tomatoes (2.6 days). In another study, short half-lives of cyantraniliprole were reported in cucumbers (2.2 days) and tomatoes (2.8 days) [23]. Indoxacarb has also been reported to degrade rapidly, with half-lives of 1.92 days in cabbage, 1.13 days in pigeon pea, and 1.12 days in okra, which results are comparable to and in agreement with those of our study [24]. Similar to the initial deposits, the degradation of pesticides with the passage of time was different. Indoxacarb dissipation was faster than that of cyantraniliprole. Such differences are likely to be due to the pesticides’ different chemical structures and physicochemical properties. The factors affecting the dissipation and half-lives of pesticides applied to crops include pesticide physicochemical properties, initial residues, morphology, crop species, frequency of application, environmental factors, and growth-dilution effects [18,20,23]. The latter two factors (environmental factors and growth-dilution effects) play vital roles in pesticide dissipation. Malhat et al. [14] and Lee et al. [23] described the dilution effect due to crop growth as one of the significant factors that causes the dissipation of pesticides in crops. In this work, although cyantraniliprole and indoxacarb were similarly applied to wild garlic, the rate of degradation of indoxacarb was faster than that of cyantraniliprole, which suggests that the variation in dissipation and half-life depended on their physicochemical characteristics.

### 3.3. Residual Behavior of Pesticides

Cyantraniliprole and indoxacarb were sprayed twice on wild garlic to evaluate the residual behavior. After spraying, the pesticide residues in wild garlic were measured at 0, 3, 7, and 14 days after spraying (Table 3 and Appendix A). The statistical analysis was carried out using a completely randomized design, and means were separated by the least significant differences (LSDs) (Table 3). However, after seven days, the residues of cyantraniliprole and indoxacarb in wild garlic were significantly decreased from 0.04 to 0.01 mg/kg and 0.14 to 0.01 mg/kg, respectively. The residual behavior of pesticides is affected by several factors [25]. In the current study, the tendency of indoxacarb residues to decrease over time was slightly faster than that of cyantraniliprole residues. In previous studies, after seven days of application, the residues of cyantraniliprole were 0.08 mg/kg in cabbage and those of indoxacarb were 4.5 mg/kg in tea, which values were higher than those determined in our study [15,16]. These higher residues may be attributed to differences in the crops’ morphological features, as garlic plants have narrow leaves and so are less likely to be covered by pesticides. At the same time, cabbage and tea have comparatively curved and broad leaves [26]. Unlike narrow leaves, foliar sprays deposit high amounts of pesticides on curved and broad leaves. Another set of factors that affect the residual behavior of pesticides are their physicochemical properties [27]. The log P value of a pesticide is an index used to predict water solubility and lipophilicity. Pesticides with a log P value greater than 1.0 have excellent penetrating characteristics [17]. In the cases of cyantraniliprole and indoxacarb, the log P values were 1.9 and 4.6, respectively, indicating their ability to penetrate the crops [28,29]. Thus, it is anticipated that the higher levels of indoxacarb residues in wild garlic are to be attributed to the high concentration of the active ingredient present in the commercial formulation and its higher log P value. The present results agreed with previous reports that pesticides’ initial residues, residual behavior, and half-lives depend on the physicochemical properties of pesticides and on crop morphology [17,23,30].

### 3.4. Pre-Harvest Interval (PHI)

The PHI is defined as the time (days) interval between the last application day of pesticides and the harvest such that the residues of pesticides decline below the MRL. Table 4 shows cyantraniliprole’s and indoxacarb’s PHI and MRL values for wild garlic. The MRL established by the Korean Ministry of Food and Drug Safety Administration (MFDS) for indoxacarb in wild garlic is 0.05 mg/kg. In contrast, the MRL for cyantraniliprole in wild garlic has yet to be established. Therefore, this study recommends an MRL value of 0.03 mg/kg for cyantraniliprole in wild garlic, following the MFDS guidelines based upon the higher value of the residual amount obtained during supervised trials. According to the current study, cyantraniliprole and indoxacarb declined below the stated MRLs within seven days between the last application and harvesting. Therefore, the safe use of cyantraniliprole 5% SC and indoxacarb 10% WP in wild garlic is recommended as two treatments seven days before harvest.

### 3.5. Safety Assessment

The TMDIs of cyantraniliprole and indoxacarb in wild garlic consumed by the Korean population are shown in Table 5. Korean Health Industry Development Institute (KHIDI) data shows that the average Korean adult (60 kg weight) consumes 0.02 g/day of wild garlic. The ADI of cyantraniliprole established by RDA is 0.057 mg/kg per body weight per day, and that of indoxacarb is 0.01 mg/kg per body weight per day. The percent ADIs of cyantraniliprole and indoxacarb calculated in this work were 0.00003% and 0.0067%, respectively. Regarding percent ADI, Chun and Kang [30] suggested that the risk of pesticides is somewhat low when the %ADI value is >10%. The respective TMDIs of cyantraniliprole and indoxacarb were 9.80% and 60.54%. The TMDI values determined in this study did not exceed 80%, which means that no potential risk to consumers could occur. Thus, due to relatively low values, cyantraniliprole and indoxacarb residues in wild garlic pose a low risk to human health [31].

## 4. Conclusions

This study evaluated the dissipations and residues of cyantraniliprole and indoxacarb insecticides in wild garlic by LC-MS/MS. Both insecticides were rapidly dissipated from wild garlic grown in greenhouse conditions. The half-lives (days) of cyantraniliprole and indoxacarb in wild garlic after seven days were 1.83 and 1.14, respectively. The indoxacarb dissipation (93%) was more rapid than that of cyantraniliprole (75%). The current work sets the PHIs of cyantraniliprole 5% EC and indoxacarb 10% WP as seven days before harvesting, with two treatments. The safety assessment in terms of TMDI revealed that wild garlic could safely be consumed following the guidelines recommended by the current investigation. The data obtained in the current work can serve to manage and regulate pesticide use to control pests in wild garlic and assist in sustainable food crop production.

## Figures and Tables

**Figure 1 toxics-11-00219-f001:**
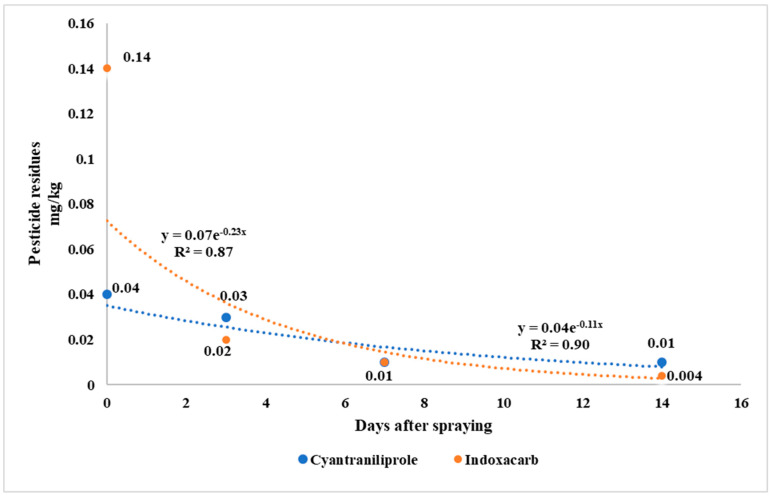
The cyantraniliprole and indoxacarb insecticide dissipation patterns (%) in wild garlic.

**Table 1 toxics-11-00219-t001:** Average recoveries (%) and LOQ of cyantraniliprole and indoxacarb in wild garlic.

Pesticides	Fortification Level (mg/kg)	(Recoveries ± RSDs ^(a)^) (%)		LOQ ^(b)^ (mg/kg)
		Intraday	Interday	
Cyantraniliprole	0.01 0.1	94.2 ± 5.9 98.7 ± 6.0	89.8 ± 7.4 93.8 ± 7.7	0.01
Indoxacarb	0.01 0.1	111.4 ± 4.2 105.0 ± 1.8	98.5 ± 5.7 95.2 ± 4.2	

^(a)^ Relative standard deviation. ^(b)^ Limit of quantitation.

**Table 2 toxics-11-00219-t002:** Cyantraniliprole and indoxacarb initial residues in wild garlic.

Pesticides	Residual Amount at 0 Day (mg/kg)	Initial Residual Amount (mg/kg)
Cyantraniliprole	0.04	0.4
Indoxacarb	0.14	0.7

**Table 3 toxics-11-00219-t003:** Residual amounts of cyantraniliprole and indoxacarb in wild garlic.

Pesticides	Days after the Last Spraying	Average Residual Amount (mg/kg)
Cyantraniliprole	0	0.04 ^B^
	3	0.03 ^CB^
	7	0.01 ^ED^
	14	0.01 ^ED^
Indoxacarb	0	0.14 ^A^
	3	0.02 ^CD^
	7	0.01 ^ED^
	14	<0.01 ^E^

Means with the same letter(s) in the same column are not significantly different (at *p* = 0.05).

**Table 4 toxics-11-00219-t004:** MRLs and PHIs of cyantraniliprole and indoxacarb for wild garlic.

Pesticides	Recommended PHI	MRL (mg/kg)
Cyantraniliprole	Two treatments 7 days before harvest	0.03 ^(a)^
Indoxacarb		0.05

^(a)^ Recommended MRL.

**Table 5 toxics-11-00219-t005:** Safety assessment of cyantraniliprole and indoxacarb in wild garlic.

Pesticides	MRL (mg/kg)	Daily Intake (g/Day)	EDI (mg/kg)	%ADI (mg/kg)	TMDI (%)
Cyantraniliprole	0.03 ^(1)^	0.02	0.1 × 10^−5^	0.3 × 10^−4^	9.80
Indoxacarb	2.0 ^(2)^		0.4 × 10^−4^	6.7 × 10^−2^	60.54

^(1)^ Recommended MRL. ^(2)^ Korean Ministry of Food and Drug Safety Administration.

## Data Availability

The data used are presented in the manuscript and are available in the Appendix A.

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
