# Peer review of "Residues and Safety Assessment of Cyantraniliprole and Indoxacarb in Wild Garlic (Allium vineale)"

_toxics, 2023, doi:10.3390/toxics11030219_

Round 1
Reviewer 1 Report
This study has done a lot of research on the degradation of cyantraniliprole and indoxacarb in Wild Garlic. This work was relatively complete and interesting. All this together gives an appropriate manuscript for the journal, but the following points should be checked:
1 the LC-MS/MS in keyword should be changed as UPLC, and also UHPLC should changed.
2 in figure1, what’s the meaning of SE
3 What is the number of measurements per data point
4 It is suggested to modify the data in Table 5 with Scientific notation
Author Response
Reviewer 1
Comments and Suggestions for Authors
This study has done a lot of research on the degradation of cyantraniliprole and indoxacarb in Wild Garlic. This work was relatively complete and interesting. All this together gives an appropriate manuscript for the journal, but the following points should be checked:
- the LC-MS/MS in keyword should be changed as UPLC, and also UHPLC should changed.
Response: Editorial change in keyword was carried out by UHPLC to UPLC as suggested.
- in figure1, what’s the meaning of SE
Answer: The ±SE represents the standard error.
- What is the number of measurements per data point?
Answer: The measurement is mg/kg (ppm). (See Fig. 1.)
- It is suggested to modify the data in Table 5 with Scientific notation
Answer: Agreed the data in table 5 were modified as recommended.

Reviewer 2 Report
General comments:
This paper demonstrated the dissipation residue and dietary risk evaluation of two insecticides in wild garlic. The extraction method was QuEChERS approach and the detection method was UPLC-MS/MS. The determination method was simple and accurate and successfully applied on the filed trials. Cyantraniliprole and indoxacarb were degraded fast in wild garlic and the dietary risk could be negligible for consumers. This study may provide some information for the safe and proper use of these two insecticides in wild garlic and other crops. I suggested that this paper could be considered for acceptance after some revisions.
Specific comments:
1. Some background information of residue researches for the two insecticides in other crops should be added to demonstrate the importance of this study.
2. Please explain the reason for selecting cyantraniliprole and indoxacarb as the target analytes.
3. Please delete the equations (1) and (2). They are meaningless.
4. Intra-day and inter-day recovery experiments are important for the method validation. Please add the inter-day recovery data.
5. Statistical analysis for the field data should be conducted to ensure the accuracy and precision of the obtained dissipation, residue and dietary risk results.
6. Please check and unify the format of references in accordance with the guidelines of “Toxics”.
Author Response
Reviewer 2
Comments and Suggestions for Authors
General comments:
This paper demonstrated the dissipation residue and dietary risk evaluation of two insecticides in wild garlic. The extraction method was QuEChERS approach and the detection method was UPLC-MS/MS. The determination method was simple and accurate and successfully applied on the filed trials. Cyantraniliprole and indoxacarb were degraded fast in wild garlic and the dietary risk could be negligible for consumers. This study may provide some information for the safe and proper use of these two insecticides in wild garlic and other crops. I suggested that this paper could be considered for acceptance after some revisions.
Specific comments:
- Some background information of residue researches for the two insecticides in other crops should be added to demonstrate the importance of this study.
Response: Agreed; more information about cyantraniliprole and indoxacarb in other crops was inserted in the introduction and results and discussion sections as suggested.
- Please explain the reason for selecting cyantraniliprole and indoxacarb as the target analytes.
Response: The reason for selecting cyantraniliprole and indoxacarb in this study was explained in the introduction section, lines 46 to 61.
- Please delete the equations (1) and (2). They are meaningless.
Response: Agreed; equations 1 and 2 were removed as recommended.
- Intra-day and inter-day recovery experiments are important for the method validation. Please add the inter-day recovery data.
Response: Agreed; inter-day recovery data were provided as suggested.
- Statistical analysis for the field data should be conducted to ensure the accuracy and precision of the obtained dissipation, residue and dietary risk results.
Response: Agreed; the statistical analysis for residual data was carried out as recommended (see table 3).
- Please check and unify the format of references in accordance with the guidelines of “Toxics”.
Response: Agreed, the references were checked as suggested.

Reviewer 3 Report
The manuscript is, in relation to its structure, in principle a modification of the previous work of the Authors [13]. Both the layout and the parts of the work remain the same. What's more – individual sentences – after adapting them to other analytes and a different matrix – are only modified by the Authors by using slightly different words. Some of these words, however, remain the same, so it is not difficult to find a similarity between these works. This applies more or less to the entire work – from the Introduction to the Conclusions. Taking this fact into account, the manuscript should be completely edited in its form.
Another problem of this work are the calibration curves presented in Figures S1 and S2. It's hard to comment, with R2 much less than 1. The arrangement of points on the graphs does not even suggest that they are supposed to create calibration curves. Thus, all calculations and results are in question and it is practically difficult to refer to them, because methodically these calibration equations should not be used with such a small correlation of data.
Moreover:
- the phrase "Residues behavior" - in the title of the work is not very correct in relation to pesticides,
- the Introduction should contain a review/reference to similar works on this subject so as to outline the background of the research (state of science) and present possible elements of the novelty of the work,
- the quoted formulas in the work – in my opinion they are unnecessary – some of them are completely basic, such as (3), and some are available in the literature, to which it would be enough to refer,
- other problems with text such as different units of concentration in Figures S1 and S2 and incorrect caption under figure S2 and so on.
Author Response
Reviewer 3
Comments and Suggestions for Authors
The manuscript is, in relation to its structure, in principle a modification of the previous work of the Authors [13]. Both the layout and the parts of the work remain the same. What's more – individual sentences – after adapting them to other analytes and a different matrix – are only modified by the Authors by using slightly different words. Some of these words, however, remain the same, so it is not difficult to find a similarity between these works. This applies more or less to the entire work – from the Introduction to the Conclusions. Taking this fact into account, the manuscript should be completely edited in its form.
Response: Agreed; the whole manuscript was revised to address the comments raised.
Another problem of this work are the calibration curves presented in Figures S1 and S2. It's hard to comment, with R2 much less than 1. The arrangement of points on the graphs does not even suggest that they are supposed to create calibration curves. Thus, all calculations and results are in question and it is practically difficult to refer to them, because methodically these calibration equations should not be used with such a small correlation of data.
Response: The calibration curve in this study was constructed following Korean ministry of food and drug safety and SANTE 11312/2021 guidelines. According to them a calibration curve having linearity of 0.99 is suitable for quantification of analytes. Further six points calibration curve was used in the study and the residues were inside the range of calibration curve.
Moreover:
- the phrase "Residues behavior" - in the title of the work is not very correct in relation to pesticides,
Response: Agreed; the title was checked and modified by removing the word “behavior” to read “Residues and Safety Assessment of Cyantraniliprole and Indoxacarb in Wild Garlic (Allium vineale)”
- the Introduction should contain a review/reference to similar works on this subject so as to outline the background of the research (state of science) and present possible elements of the novelty of the work,
Response: Agreed; the introduction section was supported by citing relevant papers as suggested.
- the quoted formulas in the work – in my opinion they are unnecessary – some of them are completely basic, such as (3), and some are available in the literature, to which it would be enough to refer,
Response: Agreed; the formulas were removed from the manuscript as suggested and instead referred to the citation.
- other problems with text such as different units of concentration in Figures S1 and S2 and incorrect caption under figure S2 and so on.
Response: The units of concentration and the caption were checked and corrected as recommended.

Reviewer 4 Report
In this work, authors detected the two pesticides of Cyantraniliprole and Indoxacarb in Wild Garlic by UHPLC-MS/MS. The residues and dissipation of Cyantraniliprole and Indoxacarb was investigated, and the Risk Assessment was conducted. This study is novel and has implications for the use of two pesticides and food safety. However, additional experiments and manuscript revisions are needed to make this work ready for publication.

Author Response
Reviewer 4
Comments and Suggestions for Authors
In this work, authors detected the two pesticides of Cyantraniliprole and Indoxacarb in Wild Garlic by UHPLC-MS/MS. The residues and dissipation of Cyantraniliprole and Indoxacarb was investigated, and the Risk Assessment was conducted. This study is novel and has implications for the use of two pesticides and food safety. However, additional experiments and manuscript revisions are needed to make this work ready for publication.
Response: The manuscript was carefully revised to address the comments raised.

Round 2
Reviewer 2 Report
Background of this paper was still not clear and no statistical analysis was conducted.
Author Response
Dear/ Editor-In chief
Subject: Manuscript ID toxics-2153035
Dear Sir,
First, we would like to thank you and the reviewers for the valuable comments, which will improve the manuscript. I am pleased to provide the author's responses to the comments raised;
The reviewer's comments and amendments were addressed in the manuscript body. References in the text and in the list were revised.
Reviewer 2
Comments and Suggestions for Authors
Background of this paper was still not clear and no statistical analysis was conducted.
Response: The background information was checked, and the statistical analysis was done as suggested (see lines 205-208 and Table 3).
Regards

Reviewer 3 Report
The work contains few corrections compared to the original version. Apart from - quite cosmetic changes - the main two problems are still visible:
1. Standard curves: The Authors, in response to the reviewer's comments, write that "According to them (guidelines) a calibration curve having linearity of 0.99 is suitable for quantification of analytes". In their work, they state that R2 = 0.18 and 0.24 (Figure S1 and S2). These values - with regard to method validation - are absolutely unacceptable. The shape of graphs on Figures S1 and S2 proves some - unknown problem with this analytical method and - one thing can be said that a linear relationship does not occur in this case. In other words, there is practically no linear correlation between peak area and ammount of pesticides. Thus, the presented data do not have much connection with the real ones.
2. This manuscript is a rework of a previous publication [13]. It has been adapted to other analytes and matrix, but unfortunately their similarity is still striking. The Authors did not introduce corrections as declared in the response to the previous review.
Author Response
Dear/ Editor-In chief
Subject: Manuscript ID toxics-2153035
Dear Sir,
First, we would like to thank you and the reviewers for the valuable comments, which will improve the manuscript. I am pleased to provide the author's responses to the comments raised;
The reviewer's comments and amendments were addressed in the manuscript body. References in the text and in the list were revised.
Reviewer 3
Comments and Suggestions for Authors
The work contains few corrections compared to the original version. Apart from - quite cosmetic changes - the main two problems are still visible:
- Standard curves: The Authors, in response to the reviewer's comments, write that "According to them (guidelines) a calibration curve having linearity of 0.99 is suitable for quantification of analytes". In their work, they state that R2 = 0.18 and 0.24 (Figure S1 and S2). These values - with regard to method validation - are absolutely unacceptable. The shape of graphs on Figures S1 and S2 proves some - unknown problem with this analytical method and - one thing can be said that a linear relationship does not occur in this case. In other words, there is practically no linear correlation between peak area and ammount of pesticides. Thus, the presented data do not have much connection with the real ones.
Response: Clarification; Line 138 to 140 clearly mentioned that the linearity of the six-point matrix matched the calibration curve for cyantraniliprole and indoxacarb was good with a correlation coefficient (R2) of > 0.99, and Table S3 and figure S1 and S2 also shows the linearity of calibration curve. The R2 value of 0.18 and 0.24 is not mentioned in the manuscript. Further the calibration curve samples were run in duplicate on top of the sequence and at the end of the sequence and then the calibration curve was constructed from the duplicate samples as shown in the example below.
- This manuscript is a rework of a previous publication [13]. It has been adapted to other analytes and matrix, but unfortunately their similarity is still striking. The Authors did not introduce corrections as declared in the response to the previous review.
Response: The manuscript was revised as suggested.
Regards

Round 3
Reviewer 2 Report
This paper could be accepted.
Author Response
Reviewer 2
Comments and Suggestions for Authors
This paper could be accepted.
The manuscript is carefully revised.

Reviewer 3 Report
Compared to the previous version, the authors have finally included the standard curves Fig S1-S2, the appearance of which is acceptable. They were made on the basis of only two repetitions for each point. This is acceptable, although it is surprising to see such a minimalistic approach to this basic validation parameter.
The question remains: what standard curves were used for the results presented in this paper: those from previous versions of the manuscript, or those from the latest version? Despite the change of simple calibration equations - the results presented in the work have not changed in relation to the original version. Can you explain it somehow?
The authors did not introduce changes related to the similarity to the publication [17]. I have already commented on this problem in previous reviews (Note 1 in Review 1 and Note 2 in Review 2).
Author Response
Reviewer 3
Comments and Suggestions for Authors
Compared to the previous version, the authors have finally included the standard curves Fig S1-S2, the appearance of which is acceptable. They were made on the basis of only two repetitions for each point. This is acceptable, although it is surprising to see such a minimalistic approach to this basic validation parameter.
The question remains: what standard curves were used for the results presented in this paper: those from previous versions of the manuscript, or those from the latest version? Despite the change of simple calibration equations - the results presented in the work have not changed in relation to the original version. Can you explain it somehow?
Response: Clarification; The only original calibration curves were presented in the original manuscript and the revised versions; therefore, the results were not changed.
The authors did not introduce changes related to the similarity to the publication [17]. I have already commented on this problem in previous reviews (Note 1 in Review 1 and Note 2 in Review 2).
Response: The manuscript is thoroughly revised as recommended to address the raised comments.

Round 4
Reviewer 3 Report
I strongly recommend further changes in the text to reduce the resemblance to [17].
Note regarding the Authors' response to the problems with the standard curve (from the previous review):
Authors response: "The only original calibration curves were presented in the original manuscript and the revised versions; therefore, the results were not changed."
Please compare:
Figure S1. Calibration curve used for quantification of cyantraniliprole in wild garlic):
Primary version of the supplementary materials:
Final version of the supplementary materials:
Has nothing really changed?
Author Response
Dear/ editor-In chief
First, we would like to thank you and the reviewers for the valuable comments, which will improve the manuscript. I am pleased to provide the author’s responses to the comments raised;
Reviewer 3:
Comments and Suggestions for Authors
I strongly recommend further changes in the text to reduce the resemblance to [17].
Response: Agreed; the manuscript is thoroughly revised as recommended.
Note regarding the Authors' response to the problems with the standard curve (from the previous review):
Authors response: "The only original calibration curves were presented in the original manuscript and the revised versions; therefore, the results were not changed."
Please compare:
Figure S1. Calibration curve used for quantification of cyantraniliprole in wild garlic):
Primary version of the supplementary materials:
Response: Bottom of Form
The data were calculated using the original calibration curve (R2 0.99); therefore, the results were not changed. The calibration curve (R2=0.1834) may have been mistakenly uploaded as a calibration curve in the supplementary materials' old version, and we apologize for this mistake. Moreover, the linearity equations of the original calibration curve are also provided in the supplementary tables in the first version.
Best regards

Round 5
Reviewer 3 Report
In the original version of supplementary materials, the authors did not provide inear equations (I have the saved original version of this file). So these explanations are, let's call it, "inaccurate." I will not once again prove something that the authors know very well.
In its current form - after a few corrections made by the Authors - the manuscript may be acceptable.